# Causal Foundation Models Perform Better without Post-treatment Variables

**Junha Ham** [* 1]   **Deokgyu Kim** [* 1]   **Doeun Kim** [* 2]   **Serjin Kim** [* 1]   **Sanghack Lee** [1]

## Abstract

Causal Foundation Models (CFMs) amortize Bayesian causal inference by pretraining on synthetic datasets, enabling zero-shot Conditional Average Treatment Effect (CATE) estimation. We study the structural bias induced when an inference-time query includes post-treatment covariates. We decompose the resulting CATE bias into three classical components — a natural indirect effect, an interaction term, and a treatment-differenced selection bias — and verify that the empirical CATE error of two representative CFMs, Do-PFN and CausalPFN, is consistent with the corresponding theoretical bounds under mediator and collider conditioning. Empirically, removing post-treatment covariates from the query yields substantial reductions in estimation error across post-treatment topologies, without retraining. As a practical alternative when the structural graph is unavailable, we introduce Treatment-Centric Local Discovery (TC-LD), a lightweight pre-inference filter that recovers most of this improvement on both fully-synthetic and semi-synthetic benchmarks.

## 1. Introduction

Recent work on Causal Foundation Models (CFMs) seeks to automate causal-effect estimation, with several approaches building on Prior-Data Fitted Networks (PFNs) (Müller et al., 2022; Ying et al., 2021; Balazadeh et al., 2025; Ma et al., 2025; Dhir et al., 2025; 2024; Hollmann et al., 2023). These models infer interventional quantities from observational context through in-context learning and support zero-shot prediction by amortizing inference over a broad class of causal mechanisms. This line of work has shown strong empirical performance on conditional average treatment effect (CATE) estimation tasks, which we also confirm

---
[*]Equal contribution  [1]Causality Lab, Seoul National University, Seoul, South Korea [2]Sogang University, Seoul, South Korea. Correspondence to: Sanghack Lee <sanghack@snu.ac.kr>.

*Proceedings of the $2^{nd}$ ICML Workshop on Foundation Models for Structured Data*, Seoul, South Korea. 2026. Copyright 2026 by the author(s).

in our benchmark experiments (Appendix G).

A key unresolved issue is whether CFMs can determine which covariates to condition on at inference time. Some approaches rely on in-context learning to recover appropriate adjustment behavior from observational context alone, without structural guidance (Robertson et al., 2025; Dhir et al., 2024). Others restrict the problem through assumptions such as strong ignorability or a fixed causal structure (Balazadeh et al., 2025; Ma et al., 2025). Our results suggest limitations in both strategies. When CFMs are queried with post-treatment covariates, they hold these covariates fixed while toggling the treatment. This induces post-treatment bias. Mediator conditioning blocks causal pathways, while collider conditioning opens spurious ones.

This failure mode can be mitigated without retraining the CFM or reconstructing a full causal graph. We introduce Treatment-Centric Local Discovery (TC-LD), a lightweight pre-inference filter that screens treatment-local problematic covariates before inference. Empirically, TC-LD recovers $85.5\%$ to $92.6\%$ of the gain achieved by oracle exclusion on our synthetic benchmark, outperforms classical causal-discovery baselines, and adds only a small computational cost.

Our contributions are as follows:

- **Theoretical insight.** We derive an exact decomposition of post-treatment bias in CFMs, showing how bad controls such as mediators and colliders distort CATE estimation.
- **Oracle analysis.** We show that removing post-treatment covariates at inference time substantially reduces CATE estimation error without retraining.
- **Practical solution.** We introduce Treatment-Centric Local Discovery (TC-LD), a lightweight method that screens treatment-local problematic covariates before inference, enabling CFMs to approach oracle-level performance in practice.

## 2. Problem Setup

### 2.1. Backgrounds on Causal Inference and PFNs

**Structural Causal Models and Interventions.** Causality is commonly formalized using Structural Causal Models (SCMs). An SCM $\psi := \langle \mathcal{G}, \{f_k\}_{k=1}^K, p(\epsilon) \rangle$ con-

sists of a DAG $\mathcal{G}$, deterministic structural assignments $x_k = f_k(\mathrm{pa}(x_k), \epsilon_k)$, and mutually independent exogenous noises with joint distribution $p(\epsilon)$. An intervention $\mathrm{do}(T = t)$ replaces the mechanism of $T$ with a constant $t$, yielding the interventional SCM $\psi_{\mathrm{do}(t)}$ and the interventional density $p(y \mid \mathrm{do}(t), x, \psi)$.

**Notation and CATE.** We use uppercase letters to denote random variables and lowercase letters for their realized values (e.g., $Z$ and $z$). The subscript obs indicates that the realization is drawn from the observational dataset $\mathcal{D}_{\mathrm{obs}}$ (e.g., $z_{\mathrm{obs}}$). Let $T \in \{0,1\}$ be a binary treatment, $C$ be pre-treatment covariates, and $Z$ be an arbitrary post-treatment covariate. For the outcome $Y$, let $Y(t,z)$ denote the potential outcome of $Y$ under the joint intervention $\mathrm{do}(T = t, Z = z)$, and let $Z(t)$ denote the potential value of $Z$ under $\mathrm{do}(T = t)$. We write $Y(t) := Y(t, Z(t))$ for the potential outcome under $\mathrm{do}(T = t)$ alone (i.e., with $Z$ left to its own potential value $Z(t)$). The Conditional Average Treatment Effect (CATE) is defined as: $\tau(c) := \mathbb{E}[Y(1) - Y(0) \mid C = c]$. Our theoretical decomposition does not rely on sequential ignorability (Imai et al., 2010).

## 2.2. Causal Foundation Models

CFMs are pretrained amortized estimators that use observational tabular data to estimate interventional quantities via in-context learning, without task-specific retraining.

**Training and Inference.** During training, a SCM $\psi$ is sampled from a prior $\pi(\psi)$, an observational context $\mathcal{D}_{\mathrm{obs}}$ is generated from $\psi$, and the model $q_\theta$ learns to map a context-query pair $(\mathcal{D}_{\mathrm{obs}}, t, x)$ to an interventional predictive distribution over $Y$. Training amortizes this Bayesian update by minimizing the expected KL divergence between the target PPD and the model prediction over sampled SCMs, contexts, and queries. We use the standard optimal-capacity idealization: on the training query distribution, $q_\theta$ recovers this PPD exactly. Throughout the theoretical analysis, we work in an *optimal-capacity limit*: the expected KL divergence is zero on the training query distribution, so $q_\theta$ recovers the true PPD for queries drawn from that distribution. This idealized assumption isolates structural bias from finite-capacity and optimization error. For CATE estimation, both Do-PFN and CausalPFN follow an S-learner inference pattern (Künzel et al., 2019). Given a supplied query feature vector, which in our analysis takes the form $X_{\mathrm{query}} = [c, z_{\mathrm{obs}}]$, the model evaluates two forward passes with $T = 1$ and $T = 0$ while holding $X_{\mathrm{query}}$ fixed. The estimated CATE is the difference between the two predictive means.

**Model Type and Scope.** End-to-end CFMs differ along three axes: prior, target, and identifiability assumptions. Do-PFN (Robertson et al., 2025) trains on broad SCM priors and targets the conditional interventional distribu-

tion (CID) $p(y \mid \mathrm{do}(t), x, \psi)$, while CausalPFN (Balazadeh et al., 2025) assumes strong ignorability and targets the conditional expected potential outcome (CEPO) $\mu_t(c; \psi) := \mathbb{E}[Y(t) \mid C = c, \psi]$. CausalFM (Ma et al., 2025) separates identification from inference, and MACE-TNP (Dhir et al., 2025) exposes no covariate conditioning slot; both fall outside the scope of our structural-bias analysis. We therefore restrict our analysis to Do-PFN and CausalPFN. Details are provided in Appendix A.

# 3. Structural Bias under Post-treatment Variables

## 3.1. Conditioning Covariates in CATE Estimation

For CATE estimation, the CFMs studied here adopt an S-learner-style formulation (See Appendix A for details). Using a query feature vector $X_{\mathrm{query}} = [c, z_{\mathrm{obs}}]$, CFMs perform two forward passes with $T = 1$ and $T = 0$ while holding $X_{\mathrm{query}}$ fixed, as in a S-learner (Künzel et al., 2019).

$$
\begin{aligned}
\hat{\tau}_{\mathrm{CFM}}(c, z_{\mathrm{obs}}) := {}& \mathbb{E}[Y(1) \mid Z(1) = z_{\mathrm{obs}}, C = c] \\
& - \mathbb{E}[Y(0) \mid Z(0) = z_{\mathrm{obs}}, C = c]
\end{aligned}
\tag{1}
$$

We define $\hat{\tau}_{\mathrm{CFM}}$ in Equation (1) as the interventional-conditional S-learner estimand approached by an optimal-capacity CFM under its training target. The structural bias of interest is $\mathrm{Bias}_{\mathrm{CFM}}(c, z_{\mathrm{obs}}) := \hat{\tau}_{\mathrm{CFM}}(c, z_{\mathrm{obs}}) - \tau(c)$.

In causal inference, conditioning on more variables does not necessarily improve estimation; validity depends on the causal role of the conditioned variables. Pre-treatment covariates $C$ may be valid adjustment variables when they block back-door paths between $T$ and $Y$. In contrast, post-treatment variables $Z$ can distort causal estimates. Because CFMs studied here condition on all query inputs, including $Z$ induces two canonical failures (Cinelli et al., 2022):

- **Mediator (Overcontrol Bias):** If $T \to Z \to Y$, fixing $Z$ blocks the causal pathway and recovers only a direct effect, not $\tau(c)$.
- **Collider (Selection Bias):** If $T \to Z \leftarrow Y$, conditioning on $Z$ opens a spurious path between $T$ and $Y$.

Extended cases, including neutral controls and bias amplification, are discussed in Appendix B.

## 3.2. Structural Bias Decomposition

We establish that querying a CFM with a post-treatment variable can structurally bias the CATE estimand. Extended formal definitions and theoretical assumptions are deferred to Appendix C.

**Proposition 3.1** (General Structural Bias Decomposition)**.** *For any nonparametric SCM and any* $z_{\mathrm{obs}} \in \mathrm{Supp}(Z \mid \mathrm{do}(t), c)$ *for both* $t \in \{0,1\}$, *the structural bias of* $\hat{\tau}_{\mathrm{CFM}}$,

*defined in Equation* (1), *admits the exact decomposition*

$$\text{Bias}_{\text{CFM}} = -\text{NIE}_Z(c) - \Delta_{\text{int}} + \Delta_{\text{sel}}, \qquad (2)$$

*where* $\text{Bias}_{\text{CFM}}$, $\Delta_{\text{int}}$, *and* $\Delta_{\text{sel}}$ *are evaluated at* $(c, z_{\text{obs}})$, $\text{NIE}_Z$ *is the natural indirect effect* (*Pearl, 2001*), $\Delta_{\text{int}} := \text{NDE}_Z - \text{CDE}_Z$, *and* $\Delta_{\text{sel}}$ *is the treatment-differenced selection bias.*

The proof is algebraic. We add and subtract the controlled direct effect, use $\tau = \text{NDE}_Z + \text{NIE}_Z$, and identify the remaining term as $\Delta_{\text{sel}}$ by definition. Formal definitions of $\text{NDE}_Z$, $\text{NIE}_Z$, and $\text{CDE}_Z$ are given in Appendix C.1, with the proof in Appendix C.2.

In the additive mediator SCM, $\Delta_{\text{sel}} = \Delta_{\text{int}} = 0$, so the bias reduces to $-\tau_i(c) = -\beta\delta$, where $\tau_i(c) := \text{NIE}_M(c)$ is the indirect effect (see Corollary C.3 and Assumption C.1). In the additive collider SCM, $\text{NIE}_Z = \Delta_{\text{int}} = 0$, so the bias reduces to $\Delta_{\text{sel}} = -\rho(\delta + \beta\alpha)$ (see Corollary C.4). Proposition 3.1 does not require the DAG or causal effects to be identifiable from observational data. Full proofs and a linear-Gaussian distributional companion comparing the true interventional posterior predictive distribution with the CFM output are given in Appendices C.2 to C.4.

**Empirical validation.** As shown in Figure 1, both CFMs exhibit growth in precision in estimation of heterogeneous effects (PEHE) consistent with the theoretical bounds. PEHE approaches $|\tau_i|$ under mediator conditioning and $|\Delta_{\text{sel}}|$ under collider conditioning, with full sweeps reported in Figures 3 and 4.

## 4. Observed Headroom: What if CFMs Infer without Post-treatment Variables?

To empirically validate Proposition 3.1 and quantify the PEHE reduction available from oracle exclusion, we compare the Naive approach against an Oracle-Exclude setting that removes ground-truth post-treatment covariates. We evaluate two CFMs (CausalPFN and Do-PFN) across three post-treatment topologies: MEDIATOR, COLLIDER, and MEDIATOR+COLLIDER. We denote the aggregate PEHE under the Naive and Oracle-Exclude adjustment sets as $\text{PEHE}_{\text{naive}}$ and $\text{PEHE}_{\text{oracle}}$, respectively, and define the

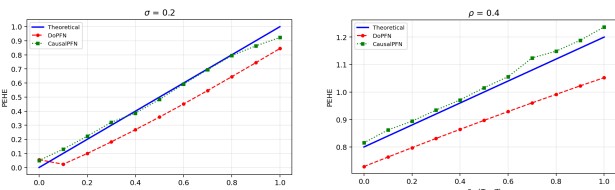

*Figure 1.* Empirical PEHE vs. theoretical bound under mediator (left, $\sigma = 0.20$) and collider conditioning (right, $\rho = 0.40$).

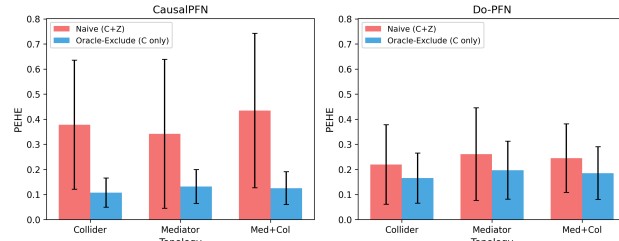

*Figure 2.* Oracle advantage by topology. PEHE under the Naive adjustment set ($C+Z$, red) vs. the Oracle-Exclude set ($C$ only, blue) for CausalPFN (left) and Do-PFN (right). Error bars denote standard deviations across configurations.

*Table 1.* Oracle vs. Random-Exclude PEHE per topology. $\Delta_{\text{oracle}}$ and $\Delta_{\text{random}}$ are relative to $\text{PEHE}_{\text{naive}}$. Negative $\Delta_{\text{random}}$ indicates that Random-Exclude performs worse than Naive.

| Model | Topology | PEHE | | | $\Delta$ | |
| | | Naive | Oracle | Random | oracle | random |
|---|---|---|---|---|---|---|
| Do-PFN | Mediator | 0.223 | 0.180 | 0.222 | +19.2% | +0.2% |
| | Collider | 0.200 | 0.149 | 0.254 | +25.5% | −26.8% |
| | M+C | 0.206 | 0.155 | 0.216 | +24.5% | −5.0% |
| CausalPFN | Mediator | 0.314 | 0.135 | 0.323 | +57.1% | −2.7% |
| | Collider | 0.427 | 0.122 | 0.464 | +71.5% | −8.7% |
| | M+C | 0.362 | 0.123 | 0.375 | +66.1% | −3.6% |

Oracle headroom as $\text{PEHE}_{\text{naive}} - \text{PEHE}_{\text{oracle}}$. The detailed experimental design is deferred to Appendix D.1.

**Oracle advantage.** As shown in Figure 2 and Table 1, removing post-treatment covariates yields substantial and consistent PEHE reductions across $n = 3,600$ samples per cell. CausalPFN achieves a 57.1%–71.5% PEHE reduction across topologies, while Do-PFN shows a stable reduction $\sim 25\%$. The Oracle advantage is driven primarily by bias reduction rather than variance. Full test statistics and effect decompositions are provided in Appendix D.1.

**Random-Exclude ablation.** To test whether the Oracle advantage is merely an artifact of reducing the feature set, we conduct a matched Random-Exclude ablation. This ablation removes the same number of randomly chosen pre-treatment covariates while retaining all mediators and colliders. Across the samples, Random-Exclude yields no gain in any of the six cases and often harms performance. As shown in Table 1, $\Delta_{\text{random}}$ ranges from $-26.8\%$ to $+0.2\%$. This indicates that the Oracle advantage is specific to removing post-treatment covariates rather than reducing the number of query features. See Appendix D.2 for detailed experiment design.

**Interior Inspection of CFMs.** To check whether CFM predictions actually rely on post-treatment features, we run two additional analyses. The first is a mediator substitution test: for each sample we swap the observed mediator $M^{\text{obs}}$ with its counterfactual $M(1 - T)$ and measure the normalized change in the predicted CATE, called indi-

*Table 2.* Mean PEHE per causal-discovery plugin method, averaged over Do-PFN and CausalPFN ($n = 1{,}440$ per method).

| CD method | PEHE | |Bias| | vs. Naive |
|---|---|---|---|
| Oracle | 0.105 | 0.072 | -55.4% |
| Ens->1 | 0.119 | 0.093 | -49.4% |
| TC-LD | 0.123 | 0.091 | -48.1% |
| Ens->2 | 0.160 | 0.130 | -32.0% |
| GES | 0.177 | 0.149 | -24.8% |
| Ens->3 | 0.195 | 0.167 | -17.3% |
| PC | 0.199 | 0.172 | -15.5% |
| FCI | 0.207 | 0.179 | -12.1% |
| Naive | 0.236 | 0.209 | – |

rect sensitivity (IS). CausalPFN reacts more to this swap than Do-PFN (mean IS 0.225 vs. 0.10). The second is a permutation-based ratio (MAR/CAR), which perturbs the post-treatment columns at inference time and measures how much the prediction shifts. Both analyses agree on two points: the models do use post-treatment information when predicting CATE, and the strength of that use is tied to the model performance, PEHE. Details are in Appendices D.3 and D.4.

## 5. Mitigating Post-treatment Bias Problem without Retraining

### 5.1. Treatment-Centric Local Discovery (TC-LD)

The Oracle-Exclude setting requires the true DAG, which is unrealistic at inference time. We introduce *Treatment-Centric Local Discovery* (TC-LD), a minimal pre-inference filter designed to recover part of this headroom; its simplicity helps isolate the role of the structural patterns documented in Sections 3 and 4. TC-LD runs two phases over each covariate $X^{(j)}$:

- **Phase 1 (or-rule screen).** Add $X^{(j)}$ to a candidate set $\mathcal{C}$ if either a Wilcoxon rank-sum test on $X^{(j)} \mid T$ or a partial correlation on $(X^{(j)}, Y \mid T)$ is significant. Phase 1 adds any covariate showing an association with either $T$ or $Y$ given $T$.
- **Phase 2 (residualization refinement).** Retain $X^{(j)}$ as pre-treatment if some subset $S \subseteq \mathcal{C} \setminus \{X^{(j)}\}$ with $|S| \leq k_{\max}$ induces conditional independence $T \perp\!\!\!\perp X^{(j)} \mid S$, as determined by a Wilcoxon rank-sum on the OLS residual. Phase 2 prunes candidates whose residual association with $T$ can be explained by conditioning on a small subset of added variables.

TC-LD has no consistency guarantee for arbitrary SCMs (failure modes detailed in Appendix D.5). Nevertheless, it performs competitively against classical baselines such as PC, FCI (Spirtes et al., 2000), and GES (Chickering, 2002) in our benchmark (Section 5.2).

### 5.2. CD Plugin Performance

We compare TC-LD against classical causal-discovery baselines, including PC, FCI (Spirtes et al., 2000), and GES (Chickering, 2002), on a benchmark spanning four DGP families (LINEAR_GAUSSIAN, LINEAR_NONGAUSSIAN, NONLINEAR_MIXED, NONLINEAR_NONGAUSSIAN) (Experiment details are in Appendix E). Each baseline serves as a pre-inference filter like TC-LD. The baseline labels covariates as pre- or post-treatment, and the CFM is evaluated on the resulting pre-treatment subset.

Within this pool, TC-LD recovers 86.7% of the Oracle headroom on average (with per-CFM ranging from 85.5% to 91.6%), while the best classical baseline (GES) recovers less than half. An ensemble that adds TC-LD to {PC, FCI, GES} recovers 89.2% of the headroom (Table 2). Per-DGP breakdown, ensemble voting analysis, and statistical tests are in Appendix E.

To complement the fully-synthetic pool, we inject mediators into IHDP (Hill, 2011) and ACIC 2016 (Dorie et al., 2019) (design in Appendix F). TC-LD detects all injected mediators (true-positive rate 1.0) across all configurations. On IHDP, no false positives occur, and TC-LD PEHE stays near the mediator-free baseline while Naive PEHE rises from 0.97 at $k = 1$ to 1.91 at $k = 8$. On ACIC 2016, Phase 2 incurs a small false-positive rate ($\approx 1\%$) in the weak-signal regime, but TC-LD still yields a net PEHE improvement over Naive at moderate-to-high mediator fractions.

**Implications for CFMs.** TC-LD does not guarantee improvement on every configuration. It produces worse PEHE than Naive on 22.4% of per-seed configurations, while the TC-LD + CD ensemble has a comparable failure rate of 24.4%. This suggests that the risk stems from imperfect post-treatment pruning rather than TC-LD-specific tests alone. TC-LD is lightweight and local. Its cost is dominated by OLS residualization and is negligible relative to a CFM forward pass.

## 6. Conclusion

We have shown that CFMs incur post-treatment bias when post-treatment covariates are included in the inference-time query set. Theoretically, we decomposed this bias into three interpretable components: the natural indirect effect, an interaction term, and treatment-differenced selection bias. Empirically, excluding post-treatment covariates at inference time yields PEHE reductions of approximately 24%–72% across topologies and models without retraining. To recover oracle headroom, we introduced TC-LD, a lightweight pre-inference filter that achieves 85.5%–92.6% of the gain by oracle exclusion on our synthetic benchmark and outperforms classical causal-discovery baselines.

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

# A. CFM Background

We briefly summarize the two end-to-end CFMs analyzed in the main body; the remaining CFMs (CausalFM, MACE-TNP) are listed for context but are outside the scope of our structural claims.

## A.1. Do-PFN: Meta-Learning over broad SCMs (Robertson et al., 2025)

Do-PFN prioritizes extreme structural flexibility by bypassing strict identifiability constraints during pre-training.

- **Identifiability and Assumptions:** Unlike traditional methods, Do-PFN does not restrict its training prior to identifiable settings. It is designed to capture *structural uncertainty* and expresses the posterior over the Markov Equivalence Class (MEC) when observational data is insufficient for point identification. Notably, it explicitly includes unobserved confounders in its prior, allowing for meta-learned adjustment under violated unconfoundedness.
- **Prior Design and Generation:**
  - **SCM Generation:** Millions of synthetic datasets are sampled. DAGs with $K \in [4, 10]$ nodes are generated via topological sorting to ensure acyclicity.
  - **Mechanisms:** Follows an Additive Noise Model (ANM): $z_k = \gamma(\sum_{l \in PA(k)} w_l z_l) + \epsilon_k$. Nonlinearity $\gamma$ is uniformly assigned from $\{x^2, \text{ReLU}(x), \tanh(x)\}$.
  - **Parameters:** Weights $w_l$ use Kaiming initialization Uniform$(-1/\sqrt{|PA(k)|}, 1/\sqrt{|PA(k)|})$. Exogenous noise $\sigma_{exo} \sim \text{Uniform}(1, 3)$, and additive noise $\sigma_\epsilon = 0.3 \cdot \beta$ where $\beta \sim \text{Beta}(1, 5)$.
- **Training Target:** Trained to predict the Conditional Interventional Distribution (CID) $p(y^{\text{in}} \mid \text{do}(t^{\text{in}}), x^{\text{in}}, \mathcal{D}_{\text{obs}})$ by NLL minimization, equivalently minimizing the expected KL divergence from true CID to the predictive $q_\theta$:

$$\min_\theta \mathbb{E}_{\psi, \mathcal{D}_{\text{obs}}} \left[ D_{\text{KL}} \left( p(y^{\text{in}} \mid x^{\text{in}}, \text{do}(t^{\text{in}}), \psi) \, \| \, q_\theta(y^{\text{in}} \mid \text{do}(t^{\text{in}}), x^{\text{in}}, \mathcal{D}_{\text{obs}}) \right) \right]$$

- **Architecture and Attention:** Transformer-based architecture optimized for tabular In-Context Learning (ICL). Employs **Row-wise Attention**, treating each observational sample as a single token to ensure permutation invariance. Features are handled via column indicators and zero-padding to accommodate varying dimensionalities. 7.3M parameters.
- **Inference Pattern:** Descendants of $T$ are identified by graph-based searches in `priors/doscm.py` and randomly included in the input feature set (`X_keys`). At inference, `predict_cate` in `base.py` toggles the treatment index while keeping the rest of the feature matrix identical (S-learner), exactly realizing Equation (1).
- **Training Environment:** Pre-trained on a single RTX 2080 GPU for 48 hours (v1) to 96 hours (v1.1) using the Adam optimizer.

## A.2. CausalPFN: Consistent Estimation under Strong Ignorability (Balazadeh et al., 2025)

CausalPFN focuses on theoretical consistency by enforcing identifiability through structural priors.

- **Identifiability and Assumptions:** Assumes **Strong Ignorability** ($Y_t \perp T|X$ and $P(T = t|X) > 0$). It is mathematically proven (Theorem 1) that as the sample size $N \to \infty$, the posterior predictive mean converges to the true Conditional Expected Potential Outcome (CEPO).
- **Prior Design and Generation:**
  - **Data Sources:** Uses 337 OpenML tables and $10^9$ feature values. Discretization of hidden neurons in a pre-trained TabPFN v1 is used to generate categorical variables.
  - **Mechanism:** Potential outcomes follow $Y_t = \mu_t(x) + \eta_t(x)\epsilon_t$. Heterogeneity is controlled by $\gamma \in [0, 1]$, where $\gamma = 1$ represents full heterogeneity. Treatment assignments are generated via RCT-like constants, linear models ($w^T x$), or MLPs.
  - **Scale:** Roughly 10,000 synthetic datasets, each with 2,048 samples.
- **Training Target:** Target is the CEPO $\mu_t(x)$, optimized via **Histogram Loss** over $[-10, 10]$ with 1,024 bins. The model minimizes cross-entropy against a narrow Gaussian distribution centered at the ground-truth $\mu_t(x)$. Consistency claim: the posterior predictive mean converges to the true CEPO as $N \to \infty$.
- **Architecture and Attention:** A 20-layer Transformer encoder with SwiGLU activations. Implements **Asymmetric Attention Masking**, ensuring query tokens attend only to context tokens. Large-scale tables ($N > 50,000$) are handled

via GBDT-based retrieval to select the most relevant context samples.

- **Inference Pattern:** Benchmarks assume $X$ is pre-treatment; at inference the feature set is duplicated across $T \in \{0, 1\}$ (S-learner). The core estimator in `causal_estimator.py` duplicates the query feature set across both treatment conditions; if a mediator is included in $X_{\text{query}}$, the model holds it constant across the two predictions, mirroring Do-PFN's S-learner pattern and failing to propagate indirect effects.
- **Training Environment:** Two-step training: (1) Predictive phase (16K context) on $4\times$ A100 for 1 week; (2) Causal phase (2K context) on $1\times$ H100 for 2 days using a schedule-free optimizer.

### A.3. Out-of-Scope Models

The following two models are included for context. Neither is subject to the query-feature misspecification analysis in the main body, for the distinct reasons explained below.

**CausalFM (Ma et al., 2025).** CausalFM is a non-end-to-end pipeline that explicitly separates identification from inference, enforcing a strict $X \to A \to Y$ structure so that $X$ is pre-treatment by construction. The data generation in `gen_standard_syn.py` follows this structure, naturally excluding post-treatment descendants from the feature set. Our analysis of query-feature misspecification does not apply directly. For reference, CausalFM uses BNN-based Cluster-DAGs (3–6 layers, 15–40 units, 50% edge dropout), a Per-Feature Transformer (10 layers, 4 heads) with a GMM head predicting CATE and CAPO, trained on a single A100 for 24 hours.

**MACE-TNP (Dhir et al., 2025).** MACE-TNP's query interface takes only an index/value tuple $(i, j, x_j)$ and produces the marginal interventional density $p(X_i \mid do(X_j{=}x_j), \mathcal{D}_{\text{obs}})$, with non-query nodes marginalized out (architecturally encoded as permutation-invariance over those nodes). Because no test-point covariate vector is exposed at inference, the query-feature misspecification we study cannot arise. For reference, MACE-TNP trains on 2.5M datasets with 2–40 nodes using an alternating sample-wise and node-wise attention mechanism, minimizing the Negative Log-Posterior Interventional Density (NLPID) via a Mixture of Gaussians output head.

## B. Extended Graphical Criteria

Building upon the graphical framework introduced by (Cinelli et al., 2022), this section provides an extended taxonomy of conditioning variables beyond the primary mediators and colliders discussed in the main text. The fundamental building blocks of directed acyclic graphs (DAGs) dictate how association flows between variables:

1. **Chains (Mediators):** $T \to Z \to Y$. Conditioning on $Z$ blocks the flow of causal association (Overcontrol Bias).

2. **Forks (Common Causes):** $T \leftarrow Z \to Y$. $Z$ is a confounder inducing spurious association. Conditioning on $Z$ blocks this back-door path.

3. **Colliders (Common Effects):** $T \to Z \leftarrow Y$. The path is closed by default. Conditioning on $Z$ induces non-causal association (Selection Bias).

While the main text focuses on the substantial bias of CFMs when conditioning on post-treatment variables, careful graphical analysis reveals other nuanced categories of covariates (Cinelli et al., 2022):

**Neutral Controls (Precision Implications).** Not all variables affect the asymptotic bias of the causal estimand. A variable $Z$ that is solely a cause of the outcome $Y$ ($Z \to Y$), without connecting to $T$, is a neutral control regarding bias. However, controlling for it absorbs variance in $Y$, potentially improving the finite-sample precision of the CATE estimate. Conversely, a variable that only causes the treatment $T$ ($Z \to T$) is also neutral for bias, but controlling for it unnecessarily reduces the variation in $T$, hurting statistical precision (Cinelli et al., 2022).

**Bias Amplification and M-Bias.** It is a common misconception that all pre-treatment variables are safe to condition on.

- **M-Bias:** If $Z$ is a pre-treatment variable that acts as a collider between two unobserved confounders ($U_1 \to Z \leftarrow U_2$, where $U_1 \to T$ and $U_2 \to Y$), conditioning on $Z$ opens the back-door path $T \leftarrow U_1 \to Z \leftarrow U_2 \to Y$, spoiling a previously unbiased estimate (Cinelli et al., 2022).

- **Bias Amplification:** If $Z$ is an instrumental variable or a strong predictor of treatment $T$ with no direct effect on $Y$, conditioning on $Z$ in the presence of unobserved confounding fails to deconfound the effect and can actively amplify existing bias in linear models (Cinelli et al., 2022).

**Case-Control Bias (Descendants of Colliders).** A variable $Z$ does not need to be a direct collider to induce selection bias. If $Z$ is a descendant of the outcome $Y$ (e.g., $T \to Y \to Z$), conditioning on $Z$ partially opens the colliding path at $Y$, thereby biasing the estimated causal effect (Cinelli et al., 2022).

## C. Theory and Proofs

### C.1. Notation and Assumptions

Let $T \in \{0, 1\}$ be the binary treatment, $C \in \mathbb{R}^d$ pre-treatment covariates, $Z \in \mathbb{R}$ a post-treatment variable (mediator $M$, collider, or mixed), and $Y \in \mathbb{R}$ the outcome. Potential outcomes: $Y(t, z)$, $Z(t)$, $Y(t) := Y(t, Z(t))$. In case-specific submodels we write $m_{\mathrm{obs}}$ for the realized value of the mediator, the natural analog of $z_{\mathrm{obs}}$ for general $Z$.

We do not assume Sequential Ignorability and permit arbitrary unmeasured confounding between $Z$ and $Y$; the conditional independence $Y(t, z) \perp\!\!\!\perp Z(t) \mid C$ is not required, and $Z$ is not restricted to lie on a directed path to $Y$.

When a CFM is queried with $X_{\mathrm{query}} = [c, z_{\mathrm{obs}}]$, its interventional prediction $\hat{\tau}_{\mathrm{CFM}}(c, z_{\mathrm{obs}})$ follows Equation (1); the structural bias is $\mathrm{Bias}_{\mathrm{CFM}}(c, z_{\mathrm{obs}}) := \hat{\tau}_{\mathrm{CFM}}(c, z_{\mathrm{obs}}) - \tau(c)$ with $\tau(c) := \mathbb{E}[Y(1) - Y(0) \mid C = c]$.

The constituent components of Proposition 3.1:

- $\mathrm{NIE}_Z(c) := \mathbb{E}[Y(1, Z(1)) - Y(1, Z(0)) \mid C = c]$,
- $\mathrm{NDE}_Z(c) := \mathbb{E}[Y(1, Z(0)) - Y(0, Z(0)) \mid C = c]$,
- $\mathrm{CDE}_Z(c, z_{\mathrm{obs}}) := \mathbb{E}[Y(1, z_{\mathrm{obs}}) - Y(0, z_{\mathrm{obs}}) \mid C = c]$,
- $\Delta_{\mathrm{int}}(c, z_{\mathrm{obs}}) := \mathrm{NDE}_Z(c) - \mathrm{CDE}_Z(c, z_{\mathrm{obs}})$,
- $\delta_{\mathrm{sel}}^{(t)}(c, z_{\mathrm{obs}}) := \mathbb{E}[Y(t) \mid Z(t) = z_{\mathrm{obs}}, C = c] - \mathbb{E}[Y(t, z_{\mathrm{obs}}) \mid C = c]$ (per-arm selection bias),
- $\Delta_{\mathrm{sel}}(c, z_{\mathrm{obs}}) := \delta_{\mathrm{sel}}^{(1)}(c, z_{\mathrm{obs}}) - \delta_{\mathrm{sel}}^{(0)}(c, z_{\mathrm{obs}})$ (treatment-differenced selection bias).

For per-$c$ quantities, we write $\mathrm{PEHE}^2(c) := \mathbb{E}_{z_{\mathrm{obs}}}\big[\big(\hat{\tau}_{\mathrm{CFM}}(c, z_{\mathrm{obs}}) - \tau(c)\big)^2\big]$ and $\mathrm{PEHE}(c) := \sqrt{\mathrm{PEHE}^2(c)}$, where the expectation is over the sampling distribution of $z_{\mathrm{obs}}$ at fixed $c$ (and analogously for $m_{\mathrm{obs}}$ in the mediator case). Aggregate PEHE is $\sqrt{\mathbb{E}_c[\mathrm{PEHE}^2(c)]}$, following Hill (2011).

**Assumption C.1** (Linear Additive SCM with a Mediator). $T = f_T(C, \epsilon_T)$, $M = \delta T + \gamma_M^\top C + \epsilon_M$, $Y = \alpha T + \beta M + \gamma_Y^\top C + \epsilon_Y$, with $\epsilon_M \perp\!\!\!\perp \epsilon_Y$ mutually independent and mean-zero. Under this structure $\mathrm{CDE}_Z = \mathrm{NDE}_Z$ (no $T \times M$ interaction), $\tau(c) = \alpha + \beta\delta$, $\tau_i(c) = \beta\delta$.

**Assumption C.2** (Linear Additive SCM with a Collider). $T = f_T(C, \epsilon_T)$, $Y = \alpha T + \gamma_Y^\top C + \epsilon_Y$, $Z = \delta T + \beta Y + \gamma_Z^\top C + \epsilon_Z$, with $\epsilon_Y \sim \mathcal{N}(0, \sigma_Y^2)$, $\epsilon_Z \sim \mathcal{N}(0, \sigma_Z^2)$ independent. $\tau(c) = \alpha$, $\tau_i(c) = 0$.

### C.2. Proof of Proposition 3.1

*Proof.* Throughout this proof, $\hat{\tau}_{\mathrm{CFM}}(c, z_{\mathrm{obs}})$ denotes the interventional S-learner object defined in Equation (1); the decomposition below is an algebraic identity under Pearl's mediation calculus and does not require Sequential Ignorability.

By definition, $\mathrm{Bias}_{\mathrm{CFM}}(c, z_{\mathrm{obs}}) = \hat{\tau}_{\mathrm{CFM}}(c, z_{\mathrm{obs}}) - \tau(c)$. Add and subtract $\mathrm{CDE}_Z(c, z_{\mathrm{obs}}) := \mathbb{E}[Y(1, z_{\mathrm{obs}}) - Y(0, z_{\mathrm{obs}}) \mid C = c]$:

$$\mathrm{Bias}_{\mathrm{CFM}}(c, z_{\mathrm{obs}}) = \big(\mathrm{CDE}_Z(c, z_{\mathrm{obs}}) - \tau(c)\big) + \big(\hat{\tau}_{\mathrm{CFM}}(c, z_{\mathrm{obs}}) - \mathrm{CDE}_Z(c, z_{\mathrm{obs}})\big).$$

By Pearl's mediation identity (Pearl, 2009; 2001), $\tau = \mathrm{NDE}_Z + \mathrm{NIE}_Z$, so the first bracket equals $-\mathrm{NIE}_Z - \Delta_{\mathrm{int}}$. By the definition of selection bias and Equation (1), the second bracket equals $\Delta_{\mathrm{sel}}$. Summing gives Equation (2). $\square$

### C.3. Mediator and Collider Corollaries

**Mediator.**

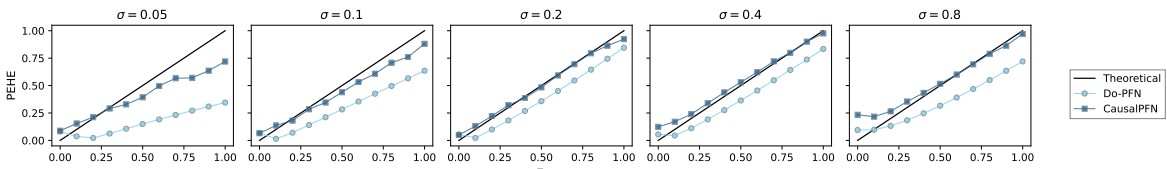

*Figure 3.* Mediator PEHE vs. $\tau_i$ across five representative noise levels $\sigma \in \{0.05, 0.10, 0.20, 0.40, 0.80\}$. Both CFMs track the theoretical bound PEHE $= |\tau_i|$ (dashed line). The full five-panel sweep is in the group repository.

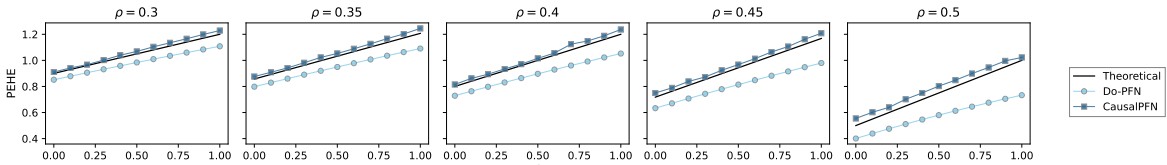

*Figure 4.* Collider PEHE vs. $\delta$ across five representative $\rho$ levels $\rho \in \{0.30, 0.35, 0.40, 0.45, 0.50\}$. The theoretical bound PEHE $= |\rho(\delta + \beta\alpha)|$ scales with $\rho$.

**Corollary C.3** (Mediator Bias). *Under Assumption C.1 (confounding-free additive SCM with $Z \equiv M$), if the CFM is queried with $X_{\text{query}} = [c, m_{\text{obs}}]$, then $\Delta_{\text{int}} = 0$ and $\Delta_{\text{sel}} = 0$, so $\hat{\tau}_{\text{CFM}}(c, m_{\text{obs}}) = \alpha$, and the structural bias equals $-\text{NIE}_Z(c) = -\tau_i(c) = -\beta\delta$.*

*Proof.* Take $Z \equiv M$ and specialize Equation (1), so $\mathbb{E}[Y(t) \mid c, m_{\text{obs}}]$ is shorthand for $\mathbb{E}[Y(t) \mid Z(t) = m_{\text{obs}}, C = c]$. Since $\epsilon_M \perp\!\!\!\perp \epsilon_Y$, conditioning on $M(t) = m_{\text{obs}}$ leaves $\mathbb{E}[\epsilon_Y \mid M(t) = m_{\text{obs}}, c] = 0$, so $\mathbb{E}[Y(t) \mid c, m_{\text{obs}}] = \alpha t + \beta m_{\text{obs}} + \gamma_Y^\top c$. Differencing across $t \in \{0, 1\}$ cancels $\beta m_{\text{obs}}$ and $\gamma_Y^\top c$, leaving $\hat{\tau}_{\text{CFM}}(c, m_{\text{obs}}) = \alpha$; subtract $\tau(c) = \alpha + \beta\delta$ for $\text{Bias}_{\text{CFM}} = -\beta\delta$. $\square$

**Collider.**

**Corollary C.4** (Collider Bias). *Under Assumption C.2 (confounding-free additive SCM), for $X_{\text{query}} = [c, z_{\text{obs}}]$, $\text{NIE}_Z = 0$, $\Delta_{\text{int}} = 0$, and the bias equals $\Delta_{\text{sel}} = -\rho(\delta + \beta\alpha)$, where $\rho := \beta\sigma_Y^2/(\beta^2\sigma_Y^2 + \sigma_Z^2)$.*

*Proof.* Since $Z$ does not enter the structural equation for $Y$ in Assumption C.2, $Y(t, z) = Y(t)$ for all $z$. Hence $\text{NIE}_Z(c) = \mathbb{E}[Y(1, Z(1)) - Y(1, Z(0)) \mid c] = 0$, and $\text{NDE}_Z(c) = \text{CDE}_Z(c, z_{\text{obs}}) = \alpha$ giving $\Delta_{\text{int}} = 0$. By Proposition 3.1, the bias reduces to $\Delta_{\text{sel}}$.

We now compute $\Delta_{\text{sel}}$. $(Y(t), Z(t))$ conditional on $C = c$ is bivariate normal with $\text{Cov}(Y(t), Z(t) \mid c) = \beta\sigma_Y^2$ and $\text{Var}(Z(t) \mid c) = \beta^2\sigma_Y^2 + \sigma_Z^2$. The linear conditional expectation is $\mathbb{E}[Y(t) \mid Z(t) = z_{\text{obs}}, c] = \mathbb{E}[Y(t) \mid c] + \rho(z_{\text{obs}} - \mathbb{E}[Z(t) \mid c])$. Substituting the SCM, $\mathbb{E}[Y(t) \mid c] = \alpha t + \gamma_Y^\top c$ and $\mathbb{E}[Z(t) \mid c] = \delta t + \beta(\alpha t + \gamma_Y^\top c) + \gamma_Z^\top c$, so $\mathbb{E}[Y(1) \mid c] - \mathbb{E}[Y(0) \mid c] = \alpha$ and $\mathbb{E}[Z(1) \mid c] - \mathbb{E}[Z(0) \mid c] = \delta + \beta\alpha$. Hence $\hat{\tau}_{\text{CFM}}(c, z_{\text{obs}}) = \alpha - \rho(\delta + \beta\alpha)$; subtracting $\tau(c) = \alpha$ gives $\text{Bias}_{\text{CFM}} = -\rho(\delta + \beta\alpha) = \Delta_{\text{sel}}$. $\square$

**PEHE bounds.** The structural bias of Corollaries C.3 and C.4 translates into closed-form PEHE limits at optimal capacity.

**Proposition C.5** (PEHE under Mediator Conditioning). *Under Assumption C.1 and Corollary C.3, $\text{PEHE}(c) \to |\tau_i(c)|$.*

*Proof.* Standard bias–variance decomposition gives $\text{PEHE}^2(c) = \text{Bias}^2 + \text{Var}_{m_{\text{obs}}}(\hat{\tau}_{\text{CFM}})$. By Corollary C.3, $\text{Bias} = -\tau_i(c)$ and $\hat{\tau}_{\text{CFM}}(c, m_{\text{obs}}) = \alpha$ does not depend on $m_{\text{obs}}$, so $\text{Var}_{m_{\text{obs}}}(\hat{\tau}_{\text{CFM}}) \to 0$ and $\text{PEHE}(c) \to |\tau_i(c)|$. $\square$

**Proposition C.6** (PEHE under Collider Conditioning). *Under Assumption C.2 and Corollary C.4, $\text{PEHE}(c) \to |\rho(\delta + \beta\alpha)|$.*

*Proof.* The argument mirrors Proposition C.5: by Corollary C.4, $\hat{\tau}_{\text{CFM}}(c, z_{\text{obs}}) = \alpha - \rho(\delta + \beta\alpha)$ is constant in $z_{\text{obs}}$, so $\text{Var}_{z_{\text{obs}}}(\hat{\tau}_{\text{CFM}}) \to 0$ and $\text{PEHE}(c) \to |\Delta_{\text{sel}}| = |\rho(\delta + \beta\alpha)|$. $\square$

**Empirical validation.** Across a noise / $\rho$ sweep, both CFMs track the theoretical bounds; see Figures 3 and 4.

## C.4. Distributional Mismatch

Beyond the point estimand $\hat{\tau}_{\text{CFM}}$, the full predictive distribution under post-treatment conditioning also strictly diverges from the true (marginalized) interventional law $P_{\text{true}}(Y) := P(Y \mid \text{do}(t), c)$. We collect the distributional definitions, lemmas, and propositions used below.

**Definitions and lemmas.**

**Definition C.7** (Kullback–Leibler Divergence). $D_{\text{KL}}(P_{\text{true}}\|Q) := \mathbb{E}_{Y \sim P_{\text{true}}}[\log(P_{\text{true}}(Y)/Q(Y))]$.

**Definition C.8** (Expected Log-Likelihood). $\text{ELL}_Q := \mathbb{E}_{Y \sim P_{\text{true}}}[\log Q(Y)]$.

**Definition C.9** (Differential Entropy). $H(P_{\text{true}}) := -\mathbb{E}_{Y \sim P_{\text{true}}}[\log P_{\text{true}}(Y)]$.

**Definition C.10** (Inference Strategies and Optimal Capacity). Let $Q_\theta$ denote the CFM's predictive distribution parameterized by $\theta$. Strategy A ($Z$ omitted from the query): $Q_A(Y) := Q_\theta(Y \mid \text{do}(t), c)$; at optimal capacity $\epsilon_A := D_{\text{KL}}(P_{\text{true}}\|Q_A) \to 0$. Strategy B (conditioned): $Q_B(Y) := Q_\theta(Y \mid \text{do}(t), z_{\text{obs}}, c)$; at optimal capacity $Q_B$ recovers $P(Y \mid \text{do}(t), z_{\text{obs}}, c)$. We write $\text{ELL}_X$ for the ELL of $Q_X$, $X \in \{A, B\}$, evaluated against $P_{\text{true}}$.

**Definition C.11** (Support). $\text{Supp}(Z \mid \text{do}(t), c)$ denotes the support of $Z$ under intervention $\text{do}(t)$ given $C = c$.

**Lemma C.12** (Gibbs' Inequality (Cover & Thomas, 2006)). $D_{\text{KL}}(P\|Q) \geq 0$, with equality iff $P = Q$ almost everywhere.

**Lemma C.13** (ELL–KL Identity). $\text{ELL}_Q = -H(P_{\text{true}}) - D_{\text{KL}}(P_{\text{true}}\|Q)$.

*Proof of Lemma C.13.* Expand $D_{\text{KL}}(P_{\text{true}}\|Q) = \mathbb{E}_{Y \sim P_{\text{true}}}[\log P_{\text{true}}(Y)] - \mathbb{E}_{Y \sim P_{\text{true}}}[\log Q(Y)] = -H(P_{\text{true}}) - \text{ELL}_Q$ and rearrange. $\square$

**Proposition C.14** (General Distributional Mismatch). *Assume $Z$ is non-deterministic under $\text{do}(t)$ and there exists a positive-measure subset $\mathcal{Z}_{\text{diff}} \subset \text{Supp}(Z \mid \text{do}(t), c)$ on which $P(Y \mid \text{do}(t), z, c) \neq P(Y \mid \text{do}(t), c)$. At optimal capacity (Definition C.10), for almost every $z_{\text{obs}} \in \mathcal{Z}_{\text{diff}}$,*

$$D_{\text{KL}}(P_{\text{true}} \| Q_B) > 0 \quad and \quad \text{ELL}_A > \text{ELL}_B.$$

*Proof.* If $D_{\text{KL}}(P_{\text{true}}\|Q_B) = 0$ held a.e. on $\text{Supp}(Z \mid \text{do}(t), c)$, then by Lemma C.12 $P(Y \mid \text{do}(t), z, c) = P(Y \mid \text{do}(t), c)$ a.e., implying interventional independence $Y(t) \perp\!\!\!\perp Z(t) \mid C = c$ — contradicting the existence of $\mathcal{Z}_{\text{diff}}$. Hence $D_{\text{KL}}(P_{\text{true}}\|Q_B) > 0$ pointwise on $\mathcal{Z}_{\text{diff}}$. By Lemma C.13, $\text{ELL}_A - \text{ELL}_B = D_{\text{KL}}(P_{\text{true}}\|Q_B) - \epsilon_A$, and at optimal capacity $\epsilon_A \to 0$, yielding $\text{ELL}_A > \text{ELL}_B$. $\square$

**Assumption C.15** (Linear Gaussian SCM with a Mediator). $T = h(C, \epsilon_T)$, $M = g(T, C) + \epsilon_M$ with $\epsilon_M \sim \mathcal{N}(0, \sigma_M^2)$ and $\sigma_M^2 > 0$; $Y = f(T, C) + \beta M + \epsilon_Y$ with $\epsilon_Y \sim \mathcal{N}(0, \sigma_Y^2)$, $\beta \neq 0$, $\epsilon_M \perp\!\!\!\perp \epsilon_Y$. Under $\text{do}(T = t)$, $M(t) \mid C = c \sim \mathcal{N}(\mu_M(t, c), \sigma_M^2)$ with $\mu_M(t, c) := g(t, c)$.

**Assumption C.16** (Linear Gaussian SCM with a Pure Collider). $T = h(C, \epsilon_T)$, $Y = f(T, C) + \epsilon_Y$ with $\epsilon_Y \sim \mathcal{N}(0, \sigma_Y^2)$; $Z = g(T, C) + \beta Y + \epsilon_Z$ with $\epsilon_Z \sim \mathcal{N}(0, \sigma_Z^2)$, $\beta \neq 0$, $\epsilon_Y \perp\!\!\!\perp \epsilon_Z$.

**Proposition C.17** (LG Mediator KL). *Under Assumption C.15,*

$$D_{\text{KL}}(P_{\text{true}} \| Q_B) = \underbrace{\frac{1}{2}\left(\frac{\beta^2 \sigma_M^2}{\sigma_Y^2} - \log\left(1 + \frac{\beta^2 \sigma_M^2}{\sigma_Y^2}\right)\right)}_{\text{variance mismatch } (>0)} + \underbrace{\frac{\beta^2(\mu_M(t,c) - m_{\text{obs}})^2}{2\sigma_Y^2}}_{\text{shift-induced bias } (\geq 0)}.$$

*Proof.* Under Assumption C.15, marginalizing $M(t)$ gives $P_{\text{true}} \sim \mathcal{N}(f(t, c) + \beta\mu_M(t, c), \beta^2\sigma_M^2 + \sigma_Y^2)$. Conditioning on $M = m_{\text{obs}}$ collapses the predictive variance: $Q_B \sim \mathcal{N}(f(t, c) + \beta m_{\text{obs}}, \sigma_Y^2)$. The univariate Gaussian KL formula yields the displayed decomposition. Since $x - \log(1 + x) > 0$ for $x > 0$, the first term is strictly positive whenever $\beta \neq 0$ and $\sigma_M^2 > 0$. $\square$

**Proposition C.18** (LG Collider KL). *Under Assumption C.16, with $\rho := \beta\sigma_Y^2/(\beta^2\sigma_Y^2 + \sigma_Z^2)$, $\mu_Z := \mathbb{E}[Z(t) \mid c]$, and $\sigma_{Q_B}^2 := \sigma_Y^2\sigma_Z^2/(\beta^2\sigma_Y^2 + \sigma_Z^2) < \sigma_Y^2$,*

$$D_{\text{KL}}(P_{\text{true}} \| Q_B) = \underbrace{\frac{1}{2}\left(\frac{\sigma_Y^2}{\sigma_{Q_B}^2} - 1 - \log\frac{\sigma_Y^2}{\sigma_{Q_B}^2}\right)}_{\text{overconfidence } (>0)} + \underbrace{\frac{\rho^2(z_{\text{obs}} - \mu_Z)^2}{2\sigma_{Q_B}^2}}_{\text{shift-induced bias } (\geq 0)}.$$

*Table 3.* Full paired-test statistics for the Oracle advantage.

| Model | Topology | $n$ | $\text{PEHE}_{\text{naive}}$ | $\text{PEHE}_{\text{oracle}}$ | $\Delta$ | $\Delta/\text{PEHE}_{\text{naive}}$ | $d$ | $p$ |
|---|---|---|---|---|---|---|---|---|
| Do-PFN | Mediator | 3600 | 0.260 | 0.197 | 0.064 | 24.4% | 0.37 | $2.5 \times 10^{-104}$ |
| | Collider | 3600 | 0.219 | 0.165 | 0.054 | 24.7% | 0.45 | $6.7 \times 10^{-145}$ |
| | Mediator+Collider | 3600 | 0.244 | 0.185 | 0.059 | 24.2% | 0.49 | $6.0 \times 10^{-169}$ |
| CausalPFN | Mediator | 3600 | 0.341 | 0.132 | 0.210 | 61.5% | 0.75 | $< 10^{-300}$ |
| | Collider | 3600 | 0.378 | 0.107 | 0.270 | 71.6% | 1.06 | $< 10^{-300}$ |
| | Mediator+Collider | 3600 | 0.434 | 0.125 | 0.309 | 71.2% | 1.04 | $< 10^{-300}$ |

*Proof.* Under Assumption C.16, $P_{\text{true}} \sim \mathcal{N}(f(t,c), \sigma_Y^2)$ since $Z$ has no causal effect on $Y$. The bivariate $(Y(t), Z(t))$ is jointly Gaussian with $\text{Cov}(Y, Z) = \beta \sigma_Y^2$ and $\text{Var}(Z) = \beta^2 \sigma_Y^2 + \sigma_Z^2$, giving the linear projection $Q_B \sim \mathcal{N}(f(t,c) + \rho(z_{\text{obs}} - \mu_Z), \sigma_{Q_B}^2)$. The Gaussian KL formula yields the decomposition. Letting $x = \sigma_Y^2 / \sigma_{Q_B}^2 - 1 > 0$, the overconfidence term equals $\frac{1}{2}(x - \log(1 + x)) > 0$. $\qquad\square$

# D. Experiment Details

## D.1. Oracle Experiment

**Design.** A paired topology benchmark varying the post-treatment structure among MEDIATOR ($T \to M \to Y$), COLLIDER ($T \to Z \leftarrow Y$), and MEDIATOR+COLLIDER (both). Each (model, configuration, seed) triple is evaluated twice under identical settings except the adjustment set: Naive $[C, Z]$ vs. Oracle-Exclude $[C]$. $n = 3{,}600$ matched pairs per (model, topology) cell. This pool generates the per-cell paired-test statistics in Table 3 and the body figure Figure 2.

**Statistical tests.** Paired $t$-test with Cohen's $d$; per-cell statistics (including exact $p$-values and effect sizes) are in Table 3.

**Effect decomposition.** Pooled over topologies, CausalPFN's Oracle gain concentrates on bias (83% drop) and indirect MAE (72% drop), with variance essentially unchanged ($< \pm 0.01$); Do-PFN shows a 31% bias drop with comparably stable variance. The marginal one-way factor ANOVA on CausalPFN identifies the fraction of post-treatment variables in the adjustment set as the single largest factor ($\eta^2 = 0.185$).

## D.2. Random-Exclude Ablation

**Design.** The Random-Exclude ablation is a symmetric control for the Oracle benchmark of Section D.1. For each (model, topology) cell, we draw $k$ pre-treatment covariates at random (matching the count $k$ that Oracle drops from the post-treatment side) and refit the CFM with that reduced feature set. If the Oracle gain were a generic dimensionality effect, Random-Exclude would reproduce it; if instead the gain is structural, dropping a confounder (pre-treatment variable) should open a back-door path and inflate PEHE. This pool generates the body table Table 1.

**Data generation.** Four DGP families (LINEAR_GAUSSIAN, LINEAR_NONGAUSSIAN, NONLINEAR_NONGAUSSIAN, NONLINEAR_MIXED), three topologies {MEDIATOR, COLLIDER, MEDIATOR+COLLIDER}, three node counts $\{5, 10, 15\}$, three post-treatment counts $\{2, 4, 6\}$, and 10 replicates per condition. After filtering combinations where $k \geq n_{\text{nodes}}$, this gives $n = 240$ rows per (model, topology) cell for mediator and collider, and $n = 280$ for mediator+collider; $n = 1{,}520$ rows in total.

## D.3. Mediator Substitution Test

The mediator substitution test draws from a separate pool of $3{,}600$ datasets (base seed $42{,}000$) restricted to mediator-only topology. The grid otherwise mirrors the Oracle design (Section D.1: same node counts, ratio levels, and direct/indirect splits).

The body (§4) defines the Indirect Sensitivity metric and reports the aggregate statistics (mean IS $0.225/0.10$, Pearson $0.619/0.448$). Conditions in which $\overline{|\hat{\tau}_{\text{obs}}|}$ falls below $10^{-12}$ are excluded from IS aggregation as numerically ill-conditioned. This eliminates fewer than $0.1\%$ of conditions. Two supplementary figures give the full visual distributions. CausalPFN's IS distribution is both higher and wider than Do-PFN's (heterogeneous mediator reliance), and the IS-vs-$|\tau_i|$ plot shows monotone dependence for both CFMs with a steeper slope for CausalPFN.

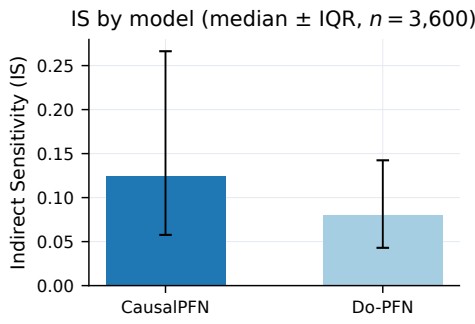

Figure 5. IS distribution across 3,600 matched samples.

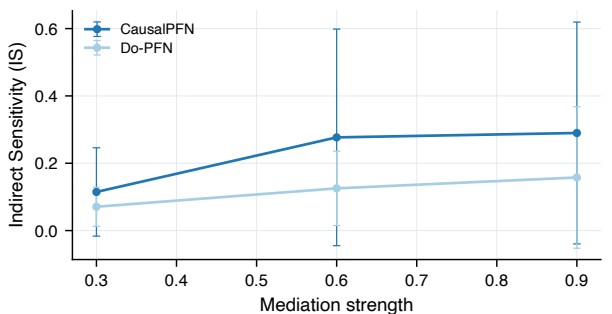

Figure 6. IS as a function of mediation strength $\alpha_M$; mean $\pm$ stddev over $\sim 1000$ (condition, rep) pairs per strength bin.

## D.4. MAR/CAR Adjustment Ratio

**Definition.** Let $\hat{\tau}_{\mathrm{orig}}$ denote the CATE prediction of a fitted CFM on the original test covariates $(C, M)$ (Naive adjustment set), and let $\hat{\tau}_{\mathrm{shuf},k}$ denote the prediction when the post-treatment columns $M$ are independently permuted at both fit and inference time (the index $k$ labels the permutation draw). The mediator adjustment ratio is

$$\widehat{\mathrm{MAR}} \;=\; \frac{1}{K}\sum_{k=1}^{K} \frac{\mathrm{RMSE}(\hat{\tau}_{\mathrm{orig}}, \hat{\tau}_{\mathrm{shuf},k})}{\mathrm{SD}(\hat{\tau}_{\mathrm{orig}})}, \qquad K = 10.$$

CAR is computed identically on collider-only topologies. The numerator measures the magnitude of the CATE shift induced by information-destroying permutations of $M$; the denominator scales by the estimator's own output variance so the ratio is comparable across conditions with different noise levels. A ratio close to $0$ means the CFM ignores $M$ in its CATE estimate; a ratio at the scale of $1$ means the CATE estimate is as variable under $M$-permutation as under its own natural spread.

**Why permute at both fit and inference.** Permuting $M$ only at inference would isolate inference-path sensitivity, but Do-PFN and CausalPFN both learn feature-level representations during fit, so a train-time-only permutation would be a different probe (fit-time robustness). Permuting at both sides measures the composite behavior of a deployed CFM under a corrupted post-treatment channel, which is the deployment scenario we care about in Section 4.

**Data generation.** We run the probe across the same four DGP families used in Section 4 (LINEAR_GAUSSIAN, LIN-EAR_NONGAUSSIAN, NONLINEAR_NONGAUSSIAN, NONLINEAR_MIXED) crossed with {MEDIATOR, COLLIDER}, three graph sizes $\{5, 10, 15\}$, three post-treatment counts $\{1, 2, 3\}$, and 10 replicates per condition, with $K{=}10$ shuffle draws and both CFMs. This gives $n_{\mathrm{conditions}} = 360$ rows per (model, topology) cell, 1,440 rows in total.

**Aggregate results.**

| Model | Topology (metric) | Mean ratio | Std | Median (IQR) |
|---|---|---|---|---|
| CausalPFN | Mediator (MAR) | 2.39 | 1.83 | 1.84 (1.02–3.22) |
| | Collider (CAR) | 3.69 | 2.49 | 3.14 (1.79–5.05) |
| Do-PFN | Mediator (MAR) | 1.44 | 1.33 | 1.02 (0.77–1.44) |
| | Collider (CAR) | 2.21 | 2.53 | 1.24 (0.92–2.09) |

The ordering CausalPFN > Do-PFN and CAR > MAR holds within every DGP family individually; for example, on LINEAR_GAUSSIAN the mean ratios are CausalPFN 5.02/2.73 (CAR/MAR) and Do-PFN 2.87/1.52, and on NONLIN-EAR_NONGAUSSIAN they are 2.59/2.07 and 1.68/1.39.

**Bias-gap correlation.** To verify that the ratio tracks bias and not just predictive variance, we re-run each (condition, replicate, model) under the Oracle-Exclude feature set, compute the per-condition bias gap $\Delta = \mathrm{PEHE}_{\mathrm{naive}} - \mathrm{PEHE}_{\mathrm{oracle}}$, and correlate it with MAR/CAR within each (model, topology) cell:

| Model | Topology | $r$ | $p$-value | $n$ |
|---|---|---|---|---|
| CausalPFN | Mediator | $+0.612$ | $2.3 \times 10^{-38}$ | 360 |
| | Collider | $+0.704$ | $3.4 \times 10^{-55}$ | 360 |
| Do-PFN | Mediator | $+0.347$ | $1.2 \times 10^{-11}$ | 360 |
| | Collider | $+0.744$ | $1.0 \times 10^{-64}$ | 360 |

All four cells show highly significant positive correlations: conditions in which the CFM responds more strongly to post-treatment permutation are also conditions in which excluding $Z$ removes more PEHE. The same pattern as the IS-vs-bias-gap correlation in Section 4 ($r=0.619$ and $0.448$) holds here under a structurally different probe (random permutation rather than counterfactual swap), confirming that MAR/CAR is tracking the same post-treatment routing that produces the Naive-vs-Oracle gap.

## D.5. TC-LD: Algorithm and Limitations

---
**Algorithm 1** Treatment-Centric Local Discovery (TC-LD)

---
1: **Input:** Dataset $\mathcal{D} = \{(T_i, X_i, Y_i)\}_{i=1}^n$, significance level $\alpha_{\text{sig}}$, max conditioning size $k_{\max}$
2: **Output:** Post-treatment index set $\mathcal{P}$
3: Initialize candidate set $\mathcal{C} \leftarrow \emptyset$
4: {*Phase 1: Screening (OR-rule)*}
5: **for** each covariate $X^{(j)}$, $j = 1, \ldots, p$ **do**
6: $\quad p_{\text{TV}} \leftarrow \text{WilcoxonRankSum}(X^{(j)} \mid T{=}1,\ X^{(j)} \mid T{=}0)$
7: $\quad p_{VY|T} \leftarrow \text{PartialCorr}(X^{(j)}, Y \mid T)$
8: $\quad$ **if** $p_{\text{TV}} < \alpha_{\text{sig}}$ **or** $p_{VY|T} < \alpha_{\text{sig}}$ **then**
9: $\quad\quad \mathcal{C} \leftarrow \mathcal{C} \cup \{j\}$
10: {*Phase 2: Conditional Refinement (subset search)*}
11: $\mathcal{P} \leftarrow \emptyset$
12: **for** each $X^{(j)} \in \mathcal{C}$ **do**
13: $\quad$ is_confounder $\leftarrow$ FALSE
14: $\quad$ **for** $s = 1, \ldots, \min(k_{\max},\ |\mathcal{C}| - 1)$ **do**
15: $\quad\quad$ **for** each $S \subseteq \mathcal{C} \setminus \{j\}$ with $|S| = s$ **do**
16: $\quad\quad\quad r^{(j)} \leftarrow X^{(j)} - \text{OLS}(X^{(j)} \text{ on } X_S)$
17: $\quad\quad\quad p' \leftarrow \text{WilcoxonRankSum}(r^{(j)} \mid T{=}1,\ r^{(j)} \mid T{=}0)$
18: $\quad\quad\quad$ **if** $p' > \alpha_{\text{sig}}$ **then**
19: $\quad\quad\quad\quad$ is_confounder $\leftarrow$ TRUE; **break**
20: $\quad\quad$ **if** is_confounder **then**
21: $\quad\quad\quad$ **break**
22: $\quad$ **if not** is_confounder **then**
23: $\quad\quad \mathcal{P} \leftarrow \mathcal{P} \cup \{j\}$
24: **return** $\mathcal{P}$

---

**OR-rule and subset search.** Phase 1 combines a Wilcoxon $p$-value $p_{\text{TV}}$ on $X^{(j)} \mid T$ with a partial-correlation $p$-value $p_{VY|T}$ on $(X^{(j)}, Y \mid T)$, admitting $X^{(j)}$ to the candidate set if *either* is significant. Phase 2 is a conservative subset search: for each candidate, we look for a conditioning set $S \subseteq \mathcal{C} \setminus \{j\}$ with $|S| \leq k_{\max}$ whose residualization eliminates the $T$–$X^{(j)}$ association; if any $S$ works, $X^{(j)}$ is classified as a confounder (retained). We fix $k_{\max} = 3$, the standard cap in constraint-based causal discovery (Spirtes et al., 2000).

**Known limitations.** (i) *Markov-equivalence barrier:* first-order parents of $T$ can be observationally indistinguishable from mediators without structural assumptions, so Phase 2 can misclassify a pre-treatment confounder. (ii) *Chain depth.* In a mediator chain $T \rightarrow M_1 \rightarrow \cdots \rightarrow M_k$, single-pass Phase 2 typically flags only $M_1$; each downstream $M_i$ is retained because its parent $M_{i-1}$ provides an in-$\mathcal{C}$ blocker. Our benchmark's post-treatment structure is predominantly shallow (single- or parallel-mediator); deeper chains would lower recovery materially. An iterative variant (flag $\rightarrow$ remove $\rightarrow$ re-apply, PC-style) would close this gap; we do not pursue it here. (iii) *Linearity in residualization.* The OLS residual step implements a linear partial-correlation CI test. A kernel-based drop-in (HSIC (Gretton et al., 2005), KCI (Zhang et al., 2011), GCM (Shah & Peters, 2020)) relaxes this at additional compute; the linear form is sufficient for our benchmark's DGP

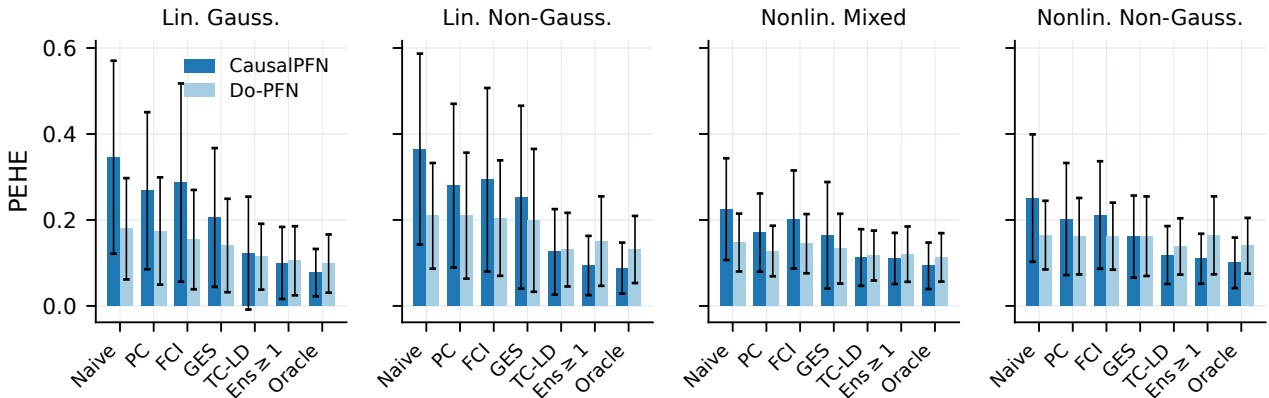

*Figure 7.* Per-DGP CD plugin breakdown. Mean PEHE per CD method across the four DGP families (LIN. GAUSS., LIN. NON-GAUSS., NONLIN. MIXED, NONLIN. NON-GAUSS.). Within each panel, paired bars show CausalPFN (dark) and Do-PFN (light) per method, with error bars showing the across-seed standard deviation. TC-LD and the ensemble dominate classical single-method baselines on every DGP.

complexity but is not essential to the design. (iv) *Power at weak mediators.* Weak mediators sharing a strong confounding parent can evade Phase 2 in finite samples.

$\alpha_{\mathrm{sig}}$-**sensitivity.** Across $\alpha_{\mathrm{sig}} \in \{0.01, 0.05, 0.10, 0.20\}$ (360 seeds per cell), median headroom recovery is stable at 0.98–0.99 for CausalPFN and essentially saturated at $\approx 1.00$ for Do-PFN; per-seed failure rates vary only modestly within each CFM (6.9%–8.6% for CausalPFN, 20.6%–25.6% for Do-PFN). We adopt $\alpha_{\mathrm{sig}} = 0.05$ as the operating point in the main text. Recovery is reported as a *median* rather than a mean because the ratio $(\mathrm{PEHE}_{\mathrm{naive}} - \mathrm{PEHE}_{\mathrm{method}})/(\mathrm{PEHE}_{\mathrm{naive}} - \mathrm{PEHE}_{\mathrm{oracle}})$ is heavy-tailed on Do-PFN cells where the Oracle headroom is near zero.

*Table 4.* TC-LD $\alpha_{\mathrm{sig}}$-sensitivity on the CD-plugin benchmark ($n$=360 seeds per cell). Median headroom recovery (row 2–3) and per-seed failure rate (row 4–5) are both reported because the failure-rate trend diverges between CFMs.

| | $\alpha$=0.01 | $\alpha$=0.05 | $\alpha$=0.10 | $\alpha$=0.20 |
|---|---|---|---|---|
| *Median headroom recovery* (Naive $\to$ 0, Oracle $\to$ 1) | | | | |
| CausalPFN | 0.978 | 0.984 | 0.989 | 0.987 |
| Do-PFN | 0.997 | 0.997 | 1.000 | 1.000 |
| *Failure rate* (TC-LD PEHE > Naive PEHE) | | | | |
| CausalPFN | 8.6% | 6.9% | 6.9% | 7.2% |
| Do-PFN | 20.6% | 21.4% | 23.9% | 25.6% |

## E. CD Plugin Benchmark: Extended Results

**Experimental setup.** The benchmark sweeps 4 DGP families (LINEAR_GAUSSIAN, LINEAR_NONGAUSSIAN, NONLIN-EAR_MIXED, NONLINEAR_NONGAUSSIAN) $\times$ 3 graph sizes $\times$ 2 mediator ratios $\times$ 3 $(\alpha_d, \alpha_i)$ pairs $\times$ 10 seeds, yielding $n$=720 configurations per (CFM, CD method) cell.

**Per-CFM split.** TC-LD (0.125) is nearly indistinguishable from the Oracle (0.121) on Do-PFN; on CausalPFN it trails Oracle (0.120 vs. 0.090) but still reduces Naive PEHE (0.297) by over $\sim 60\%$.

**Per-DGP recovery of the Oracle headroom.** TC-LD: 82.4% (LINEAR_GAUSSIAN), 89.5% (LINEAR_NONGAUSSIAN), 85.8% (NONLINEAR_MIXED), 90.6% (NONLINEAR_NONGAUSSIAN). Per-CFM recovery: 91.6% (Do-PFN), 85.5% (CausalPFN) for TC-LD, vs. 24.8–48.8% for GES. The full per-DGP breakdown is shown in Figure 7.

**Ensemble analysis (Ens-3 vs. Ens-4).** To isolate TC-LD's contribution to the 4-method ensemble Ens-$\geq$1, we paired-compare Ens-3 (majority vote over {PC, FCI, GES}) against Ens-4 (= Ens-3 + TC-LD). Adding TC-LD is statistically significant ($p \ll 10^{-8}$) across all voting thresholds; the largest gain is on CausalPFN at the $\geq$1-vote threshold (37.2% PEHE reduction; Table 5).

*Table 5.* Ens-3 vs. Ens-4 paired comparison, $n = 720$ paired seeds per row.

| FM | Threshold | $n$ | Ens-3 | Ens-4 | $\Delta$ | Rel.% | $p$ |
|---|---|---|---|---|---|---|---|
| Do-PFN | $\geq 1$ | 720 | 0.158 | 0.135 | +0.023 | +14.3% | 1.1e-10 |
| | $\geq 2$ | 720 | 0.166 | 0.149 | +0.017 | +10.1% | 2.7e-09 |
| | $\geq 3$ | 720 | 0.172 | 0.160 | +0.011 | +6.4% | 3.6e-09 |
| CausalPFN | $\geq 1$ | 720 | 0.165 | 0.104 | +0.061 | +37.2% | 2.2e-31 |
| | $\geq 2$ | 720 | 0.232 | 0.172 | +0.060 | +25.9% | 7.4e-35 |
| | $\geq 3$ | 720 | 0.276 | 0.230 | +0.046 | +16.8% | 1.7e-28 |

**Failure-rate breakdown.** TC-LD produces higher PEHE than Naive on $22.4\%$ of per-seed configurations; Ens-$\geq 1$ on $24.4\%$. These failures cluster on small-graph configurations (the MEC barrier is most acute there).

Table 6 disaggregates the failure rate by graph size $\times$ mediator ratio. The aggregate over this 2D grid ($14.2\%$) is computed on the $\alpha$-sweep seed pool (same source as Table 4) and so differs in magnitude from the main-text $22.4\%$ computed on the full CD-plugin benchmark pool; the structural pattern, however, is the same. Failures concentrate on small graphs ($n < 6$: $13.3\%$ at both ratios) and on mid-size graphs with higher mediator density ($6 \leq n < 11$, ratio 0.50: $17.5\%$); the one regime where TC-LD is consistently safe is large graphs with denser mediators ($n \geq 11$, ratio 0.50: $4.2\%$). This pattern is consistent with the MEC barrier and finite-sample power limitations flagged in Section D.5.

*Table 6.* TC-LD failure rate (TC-LD PEHE > Naive PEHE) by graph size $\times$ mediator ratio, pooled over CausalPFN and Do-PFN (120 seeds per cell, $\alpha_{\text{sig}}{=}0.05$).

| Graph size | mediator ratio=0.30 | mediator ratio=0.50 |
|---|---|---|
| $n < 6$ | 13.3% | 13.3% |
| $6 \leq n < 11$ | 15.8% | 17.5% |
| $n \geq 11$ | 20.8% | 4.2% |
| Aggregate | | 14.2% |

**Wall-clock cost (deferred).** A direct wall-clock comparison of TC-LD vs. PC/FCI/GES at $(n, p) \in \{1\text{k}, 10\text{k}\} \times \{10, 50\}$ is deferred to the conference-length follow-up. Analytically, the TC-LD screen runs in $\mathcal{O}(p)$ time for Phase 1 (two univariate tests per covariate) and $\mathcal{O}\big(p \cdot \sum_{s=1}^{k_{\max}} \binom{|\mathcal{C}|}{s}\big)$ for Phase 2 (subset search capped at $k_{\max}{=}3$, with $|\mathcal{C}|$ typically $\ll p$ after screening). On the benchmark runs that underlie Table 2 ($n{=}500$, $p \leq 20$), the TC-LD screen itself was empirically sub-second; the dominant deployment cost remains the CFM forward pass, not the screen.

## F. Semi-Synthetic Mediator Injection

We inject $k$ parallel synthetic mediators into IHDP and ACIC 2016. Each mediator follows $M_j = \gamma T + \sigma \varepsilon_j$ with independent noise, and the training outcome is additively shifted:

$$Y_{\text{new}} = Y_{\text{orig}} + \delta \sum_j M_j, \qquad \tau_{\text{new}} = \tau_{\text{orig}} + \delta \, k \, \gamma.$$

We sweep $\gamma \in \{0.3, 1.0\}$, $\delta \in \{0.3, 1.0\}$, $\sigma = 0.2$, and scale $k$ as a fraction of the covariate dimension $d$: $k \in \{1, 3, 5, 8\}$ for IHDP ($d{=}25$; 4%–32%) and $k \in \{2, 6, 11, 17\}$ for ACIC 2016 ($d{=}55$; 4%–31%), over 10 tables per benchmark with both Do-PFN and CausalPFN.

TC-LD detects every injected mediator in all configurations (Table 7). On IHDP no pre-treatment covariate is falsely flagged, and TC-LD PEHE tracks the mediator-free baseline closely while Naive PEHE grows linearly with $k$ (Figure 8). On ACIC 2016, Phase 2 misclassifies a small number of $T$-correlated pre-treatment columns (FP rate 0.9%–1.5% across $k$). This cost is concentrated in the weak-signal regime ($\gamma{=}0.3$), where the $T{\rightarrow}M$ association is comparable to the $T$-correlation of some genuine confounders; when $\gamma{=}1.0$ the mediator signal is well-separated and the penalty is negligible (+0.01). As $k$ grows, the linearly increasing Naive PEHE outweighs this roughly constant FP cost: at $k{=}17$ (31% of $d$) CausalPFN Naive PEHE reaches 3.06 while TC-LD stays at 1.18.

*Table 8.* PEHE (lower is better) across standard pre-treatment-only tabular CATE benchmarks. Bold: best; underline: second. General context; not a test of the TC-LD claim.

| Model | IHDP | ACIC 2016 | Lalonde CPS | Lalonde PSID |
|---|---|---|---|---|
| CausalFM | **0.497 ± 0.322** | **0.392 ± 0.081** | 9,060 ± 240 | **13,844 ± 2,057** |
| CausalPFN | 0.580 ± 0.702 | 0.921 ± 0.355 | **8,956 ± 209** | 14,402 ± 1,984 |
| Do-PFN | 6.065 ± 8.989 | 4.114 ± 1.733 | 12,008 ± 279 | 20,584 ± 1,349 |
| T-Learner | 2.242 ± 3.525 | 1.294 ± 0.402 | 9,214 ± 285 | 14,540 ± 1,814 |
| S-Learner | 3.221 ± 5.575 | 1.169 ± 0.433 | 12,637 ± 361 | 21,926 ± 1,370 |
| X-Learner | 3.296 ± 5.025 | 1.001 ± 0.524 | 11,964 ± 1,254 | 20,467 ± 1,909 |
| DA-Learner | 10.374 ± 16.934 | 1.276 ± 0.694 | 4,364,457 ± 2,656,526 | 3,387,038 ± 4,358,041 |
| DR-Learner | 38.073 ± 49.504 | 2.116 ± 0.516 | 16,949,205 ± 11,460,787 | 11,144,006 ± 11,963,749 |
| GRF | 3.747 ± 6.080 | 1.189 ± 0.769 | 12,765 ± 646 | 27,511 ± 8,392 |
| Causal Forest DML | 4.068 ± 6.507 | 1.340 ± 0.830 | 2,515,774 ± 2,167,071 | 164,356 ± 164,980 |

*Table 7.* TC-LD detection performance on injected mediators. TP rate: fraction of mediator columns correctly flagged as post-treatment. FP rate: fraction of pre-treatment columns incorrectly flagged. FP/cell: average number of pre-treatment columns falsely dropped per cell.

| Dataset | $k$ | $k/d$ | TP rate | FP rate | FP/cell |
|---|---|---|---|---|---|
| IHDP | 1 | 4% | 1.0 | 0.0% | 0.0 |
| | 3 | 12% | 1.0 | 0.0% | 0.0 |
| | 5 | 20% | 1.0 | 0.0% | 0.0 |
| | 8 | 32% | 1.0 | 0.0% | 0.0 |
| ACIC 2016 | 2 | 4% | 1.0 | 1.5% | 0.9 |
| | 6 | 11% | 1.0 | 1.0% | 0.6 |
| | 11 | 20% | 1.0 | 0.9% | 0.5 |
| | 17 | 31% | 1.0 | 0.9% | 0.5 |

*Figure 8.* PEHE vs. number of injected mediators $k$ on IHDP ($d$=25, left) and ACIC 2016 ($d$=55, right), faceted by CFM. Naive PEHE rises linearly with $k$; TC-LD PEHE stays close to the mediator-free baseline on IHDP and converges toward it on ACIC 2016 as the mediator fraction grows.

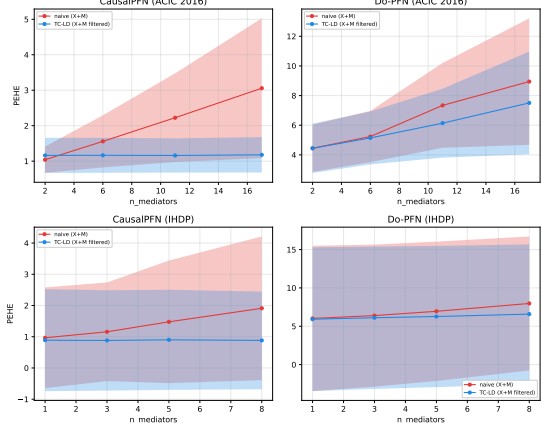

# G. Overall Performance Benchmarks

The standard tabular CATE benchmarks (IHDP (Hill, 2011), ACIC 2016 (Dorie et al., 2019), Lalonde CPS/PSID (LaLonde, 1986)) contain pre-treatment covariates only; the post-treatment failure mode studied in this paper does not arise on these datasets. Additional baselines include metalearners (T-, S-, and X-Learner) (Künzel et al., 2019), DA- and DR-Learner (Kennedy, 2023), GRF (Athey et al., 2019), and Causal Forest DML (Chernozhukov et al., 2018). The experimental setup and hyperparameters follow Balazadeh et al. (2025).

