# OpenReview forum: "Causal Foundation Models Perform Better without Post-treatment Variables"
_ICML.cc/2026/Workshop/FMSD — FMSD @ ICML 2026 Poster_

### Official Review · Reviewer_mWpW · 2026-05-19
**Good paper, but scope should be clearer**

**Rating:** 7
**Confidence:** 4

**Review:**

Summary:
The paper studies post-treatment covariate bias in Do-PFN and CausalPFN. The main issue is that these models can hold post-treatment variables fixed while changing treatment, which creates mediator/collider bias. The paper also proposes TC-LD as a simple pre-inference filter.

Strengths:
The problem is important and very relevant to the workshop. The causal explanation is clear. The oracle exclusion result is convincing, and the random-exclude ablation is a good control. TC-LD is also a useful practical addition, especially with comparisons to PC, FCI, and GES.

Areas for Improvement:
The main issue is scope. The paper often says “CFMs” broadly, but the analysis is mainly about Do-PFN and CausalPFN, or models with a similar S-learner-style query interface. This should be stated more carefully.

TC-LD is also a heuristic. Since the paper notes that it has no general guarantee and can hurt in some settings, this limitation should be more visible.

Detailed Comments:
Please clarify when TC-LD is expected to fail. Real settings may have weak effects, correlated covariates, nonlinear dependencies, or longer mediator chains, which may be harder than the current synthetic/semi-synthetic checks.

Justification of Score:
I think this is a good workshop paper. The core failure mode is important, the experiments have good controls, and the proposed fix is practical. My concerns are mainly about scope and TC-LD limitations.

---

### Official Review · Reviewer_8rjD · 2026-05-19
**A Simple Failure Mode of CFMs**

**Rating:** 6
**Confidence:** 4

**Review:**

# Summary
This paper studies a failure mode of causal foundation models (CFMs), e.g. Do-PFN and CausalPFN. When post-treatment variables are included in the query, the model toggles treatment while holding those variables fixed, causing bad CATE estimates. The authors decompose this error into indirect-effect, interaction, and selection-bias terms, show that removing post-treatment variables substantially improves PEHE, and propose TC-LD as a lightweight pre-inference filter to approximate oracle removal.

# Strengths
The paper found out that for causal foundation models, they do not know which variables are valid to condition on. The theoretical decomposition is clean and connects CFM behavior to classical mediator and collider bias. In the experiment section, the oracle removal improves PEHE substantially, while random feature removal does not, showing the gain is real.

# Areas for Improvement
The main causal insight is already obvious. It is already known that the post-treatment variables should not be controlled for when estimating total effects. The novelty is mainly in showing that CFMs still suffer from this issue. The paper should position this more clearly.

# Detailed Comments
The paper would also benefit from a simple worked example to demonstrate how the method works.

Justification of Score
I would lean weak accept. The paper is relevant and well-written but the core principle is already well-known.

---

### Official Review · Reviewer_Kp6s · 2026-05-22
**A good paper for the structural bias**

**Rating:** 7
**Confidence:** 3

**Review:**

This paper studies the structural bias induced when an inference-time query includes post-treatment covariates. It decomposes the resulting CATE bias into three classical components, including a natural indirect effect, an interaction term, and a treatment-differenced selection bias. It further verifies that the empirical CATE error of two representative CFMs, Do-PFN and CausalPFN, is consistent with the corresponding theoretical bounds under mediator and collider conditioning. It also shows how to mitigate it without retraining the model.


Pros:
1. The theoretical decomposition is elegant and sound.
2. The empirical studies sound good to me.

Cons:
1.  Most experiments are based on synthetic or semi-synthetic datasets. Although this is common in the study of causal inference, validation based on real-world datasets would make the empirical studies more convincing.